# Animal Models in Microbeam Radiation Therapy: A Scoping Review

**DOI:** 10.3390/cancers12030527

**Published:** 2020-02-25

**Authors:** Cristian Fernandez-Palomo, Jennifer Fazzari, Verdiana Trappetti, Lloyd Smyth, Heidrun Janka, Jean Laissue, Valentin Djonov

**Affiliations:** 1Institute of Anatomy, University of Bern, 3012 Bern, Switzerland; cristian.fernandez@ana.unibe.ch (C.F.-P.); jennifer.fazzari@ana.unibe.ch (J.F.); verdiana.trappetti@ana.unibe.ch (V.T.); jean-albert.laissue@pathology.unibe.ch (J.L.); 2Department of Obstetrics & Gynaecology, University of Melbourne, 3057 Parkville, Australia; lloyd.smyth@unimelb.edu.au; 3Medical Library, University Library Bern, University of Bern, 3012 Bern, Switzerland; heidrun.janka@ub.unibe.ch

**Keywords:** microbeam radiation therapy, scoping review, animal models, rat, mouse, synchrotron, spatial fractionation, radiotherapy, oncology

## Abstract

Background: Microbeam Radiation Therapy (MRT) is an innovative approach in radiation oncology where a collimator subdivides the homogeneous radiation field into an array of co-planar, high-dose beams which are tens of micrometres wide and separated by a few hundred micrometres. Objective: This scoping review was conducted to map the available evidence and provide a comprehensive overview of the similarities, differences, and outcomes of all experiments that have employed animal models in MRT. Methods: We considered articles that employed animal models for the purpose of studying the effects of MRT. We searched in seven databases for published and unpublished literature. Two independent reviewers screened citations for inclusion. Data extraction was done by three reviewers. Results: After screening 5688 citations and 159 full-text papers, 95 articles were included, of which 72 were experimental articles. Here we present the animal models and pre-clinical radiation parameters employed in the existing MRT literature according to their use in cancer treatment, non-neoplastic diseases, or normal tissue studies. Conclusions: The study of MRT is concentrated in brain-related diseases performed mostly in rat models. An appropriate comparison between MRT and conventional radiotherapy (instead of synchrotron broad beam) is needed. Recommendations are provided for future studies involving MRT.

## 1. Introduction

### 1.1. Background

In our society, cancer is an increasingly prevalent and therapeutically challenging disease. Of all cancer patients treated curatively, half will receive radiation therapy [1]. However, despite these efforts, more than a quarter of a million people die from cancer in the EU annually [2]. The main factor limiting conventional radiotherapy (RT) is the toxicity induced in vital surrounding healthy tissues, which constrains dose-escalation and effective tumour control [3]. Synchrotron X-ray microbeams have been recognized as a unique tool to overcome this limitation [4], eliciting scientific interest across the world in a radio-therapeutic application known as Microbeam Radiation Therapy (MRT). MRT consists of a spatially-modulated, co-planar array of low energy X-rays delivered to tumours [5]. These spatially fractionated beams exploit the dose-volume effect for extraordinary normal tissue tolerance [6] and trigger a cascade of biological effects that greatly improve tumour control [7].

### 1.2. Rationale

In the last few decades, MRT research has expanded to include several synchrotron and non-synchrotron sources around the world. The progress, however, has been relatively slow. A significant challenge for MRT is the wide range of modifiable radiation parameters which can profoundly impact its radiobiological effects, and ultimately, treatment outcomes. Research groups have explored these radiation parameters using various rodent and non-rodent animal models. However, there remains no clear consensus within the community on the optimal MRT parameters and strategy for cancer therapy, delaying translation to human clinical trials. Comprehensively evaluating the existing literature is essential to unifying these pre-clinical data and strategically promoting future research to support clinical translation. 

### 1.3. Objective

The objective of this scoping review is to map the available evidence and provide an overview of the similarities, differences, and outcomes of all experiments that have employed animal models in MRT. The aim is to inform the community about progress made in the field of MRT and to identify existing gaps in current research. 

## 2. Methods

### 2.1. Protocol and Registration

Our protocol was developed according to the scoping review methodological framework proposed by the Joanna Briggs Institute [8], and refined by two methodological papers on how to conduct scoping reviews published by Peters at al. [9] and Tricco et al. [10]. The draft protocol was revised by the research team of the Institute of Anatomy of the University of Bern and by the information specialist of the University of Bern’s Medical Library. The protocol could not be registered in the International Prospective Register of Systematic Reviews (PROSPERO), as this is not yet an option for scoping reviews. This scoping review was written according to the Preferred Reporting Items for Systematic Reviews and Meta-Analyses (PRISMA)-update for Scoping Reviews [11].

### 2.2. Eligibility Criteria 

The review considered articles that employed animal models with the purpose of studying the normal tissue or therapeutic effects of MRT. We included all papers that employed MRT as (i) an array of co-planar microbeams, (ii) from any radiation source (synchrotron and non-synchrotron) and quality (photons, protons, and others), (iii) that were applied to animals. We defined MRT as an array of co-planar microbeams of widths equal to or lower than 100 µm, which are spaced by a few hundred micrometres within the array. This definition of MRT has been used by other research groups, and it is becoming the standard in the field [5,12,13,14,15]. MRT is also described as spatial fractionation of the radiation beam, which is the main difference with conventional radiotherapy (Figure 1). We excluded papers that employed minibeams because their width exceeds 100 µm. We also excluded publications that employed a single microbeam and a microbeam scanning technique because these approaches do not qualify as spatial fractionation. 

### 2.3. Information Sources 

Literature searches were performed by an information specialist (HJ) in the following information sources: Medline (Ovid) (incl. Epub Ahead of Print, In-Process and Other Non-Indexed Citations, Medline Daily and Ovid Medline Versions (1946–October 02, 2019)Embase (Ovid) (1947–October 02, 2019)Cochrane Library (1996–October 02, 2019)Scopus (1788–October 02, 2019)ICTRP Trial Register (WHO)ClinicalTrials.govLivivo Search portal

### 2.4. Search Strategy 

An initial search strategy in Medline was drafted and tested against a list of core references to see if they were included in the search results. After refinement and consultation with the research team, search strategies were set up for each information source, based on database-specific, controlled vocabulary and text words. No limits have been applied in any database considering study types, languages, publication years, etc. Beside standard bibliographic databases for medical journals like Medline, Embase and the Cochrane Library, two interdisciplinary databases, Scopus and Livivo, have been selected to search for other publications types than journal articles (e.g., conference proceedings, book series, publications from university repositories). All searches were run on October 3rd, 2019. In addition to electronic database searching, bibliographies and reference lists from relevant publications were checked.

For our search strategy, we decided not to focus on any particular animal species or animal models. This allowed for maximum sensitivity in the search and ensured that publications on all organisms on which MRT had been applied were included in our search results. Therefore, our search strategy focused only on the concepts of “microbeam radiation therapy”, “synchrotron x-ray microbeams” in combination with “radiotherapy”, “therapy” or similar terms. The detailed search strategies are presented in Appendix A.

### 2.5. Selection of Sources of Evidence 

All identified citations were collated, loaded into EndNote and duplicates were removed. The collection was then uploaded into the software Rayyan [16]. Two independent reviewers performed the first level screening of titles and abstracts against the inclusion criteria. Two reviewers also performed the second level of full-text screening.

### 2.6. Data Charting and Synthesis of Results

Three reviewers determined the variables to extract. The studies were charted jointly by using Microsoft Excel Online. We grouped the studies into experimental articles, reviews, clinical perspectives, reports, commentaries, and Letters to the Editor. Reviews, commentaries, and clinical perspectives were used to screen for possibly missing citations. The experimental articles and reports were charted.

## 3. Results

### 3.1. Literature Selection

After duplicates were removed, a total of 5688 records were identified (Figure 2). Based on title and abstract screening, 5534 citations were excluded, and 159 potentially relevant papers were assessed for full-text eligibility, which includes 5 additional records identified through reference scanning. After the full-text screening, 64 articles were further excluded, and 95 studies were finally included in the review. The list of included references can be found in the Appendix A.

### 3.2. Study Characteristics

The 95 papers studying animal models in MRT were disseminated between 1994 and 2019 (Figure 3A). The majority of the articles have been published in the last ten years, with the highest publication record (11 papers) in 2015. From the 72 experimental articles, the most commonly used animal models have been Rats (59.7%) and Mice (33.3%) with only a few studies using other animal models (Figure 3B).

### 3.3. Technical Parameters

In the reviewed literature, the MRT array was created via two mechanisms. By performing several consecutive irradiations (with horizontal translation) using a single-slit collimator, to generate an array of co-planar microbeams at the target. Alternatively, by using a multi-slit collimator that generates an array of co-planar microbeams simultaneously. 

Across the included literature, the width of the microbeams ranged from 20 to 100 µm, with spacing varying from 50 up to 500µm (See Table 1; Table 2). 

MRT can be delivered as a single or multiple array(s). When multiple arrays are delivered, the irradiation geometry can be subclassified according to the trajectory of delivery and the anatomical plane. 

Trajectory of delivery: if two or more arrays were delivered in the same trajectory or direction, they were called unidirectional. However, if the arrays were delivered in two or more trajectories, they were referred to as bidirectional or multidirectional.Anatomical plane: if two or more MRT arrays were delivered in the same anatomical plane they were also called co-planar. When two co-planar arrays were delivered without overlap of the peak-dose regions, they were referred to as interlaced. Delivery in different planes was referred to as cross-planar where irradiation with a single unidirectional MRT array was followed by a rotation of the animal and a second (or third) MRT array was delivered (creating a grid-like pattern in the target volume). When arrays in this manner intersect at 90°, they are orthogonal (irrespective of the anatomical plane or trajectory).

The use of the word cross-fired however, was more general. It described two or more arrays, but it did not inform consistently about the trajectory or the anatomical plane of the arrays.

Peak-doses ranged from 3.9 Gy [17] up to 10000 Gy [18], however, the depth at which they were measured varied across the publications from 0.3 mm (skin-entry or entrance) to several millimetres in depth depending on the location of the target. Valley-doses were also reported at different depths, the lowest was 0.5 Gy at skin-entry [19], and the highest was 38.5 Gy at 5 mm depth [20]. Additionally, the peak-to-valley dose ratio (PVDR) was described as an important metric related to the therapeutic index of MRT [21].

Dose-rate was reported in only 27 out of 72 experimental articles and varied greatly from 0.3 to 16000 Gy/s. This large variation is relevant because dose-rate, together with the peak-dose, will determine whether MRT will also have a FLASH effect. The FLASH effect is defined as the delivery of radiation at ultra-high dose rates, which allows normal tissue tolerance levels to be exceeded [22,23,24]. This is achieved when the whole dose of radiation (the peak-dose in the case of MRT) is delivered in less than 200 milliseconds [24]. Therefore, not all MRT sources will be able to have a FLASH effect since, for example, the delivery of a peak-dose of 400 Gy requires a dose rate of 2000 Gy/s as a minimum to be delivered in 200 milliseconds.

### 3.4. Animal Models in MRT

Animal models are divided into their use for investigating tumour control, non-neoplastic diseases and normal tissue effects following MRT.

#### 3.4.1. MRT for Cancer 

Table 1 reports all the studies that involve the use of MRT to treat tumour-bearing animals and summarises key radiation parameters which are associated with tumour control. 

#### 3.4.2. Glioma 

The most commonly employed cancer models in MRT research were brain tumours. The first animal model used was syngeneic 9L gliosarcoma (9LGS) implanted in rats. In the early nineties, at the National Synchrotron Light Source of the Brookhaven National Laboratory, Laissue, Slatkin and colleagues [25,26,27,28] showed that MRT (unidirectional and bidirectional) delayed tumour growth of intracerebral 9LGS. Since then, 9LGS cells in the Fisher 344 strain rat model has been widely used to test several MRT configurations and doses. 

In 2002, Dilmanian and colleagues [15] studied different unidirectional arrays on the same 9LGS model. The array had a constant microbeam width (27 µm), three spacings (50 µm, 75 µm and 100 µm), and two peak-doses (150 Gy and 500 Gy). All MRT conditions improved survival as compared to the control group.

In addition, several studies conducted by Bouchet and colleagues [20,29,30], using the 9LGS model, investigated the molecular mechanisms behind the MRT efficacy in comparison to unirradiated controls.

A comparison between a homogenous broad beam (BB) of radiation delivering 17.9 Gy to the tumour with one MRT array - formed by microbeams of 50 µm width and centre-to-centre spacing of 200 µm (hereafter defined as “50 µm; 200 ctc”) - delivering 200 Gy peak-doses to the tumour revealed that the survival for the animals of both groups was similar [31]. However, when the deposited peak-dose in the MRT array was increased to 400 Gy the survival of the gliosarcoma-bearing rats was significantly increased compared to the BB irradiated animals [31].

Several studies have shown that delivering multiple MRT arrays increased the survival of 9LGS-bearing rats significantly compared with unirradiated controls [32], with characteristic induction of hypoxia at the tumour site [33]. Serduc et al. [34] employed bidirectional MRT arrays to compare the efficacy of three different microbeams widths (25, 50 and 75 µm), each delivered as two orthogonally cross-planar intersecting arrays. In all the MRT configurations the valley-dose was kept constant (at 18 Gy each) by fixing the spacing (211 ctc), and only the peak-dose and the microbeam widths were modified. Thus, microbeams of 25, 50, and 75 µm of width, had peak-dose of 860 Gy, 480 Gy, and 320 Gy per array, respectively. All MRT groups delayed the 9LGS tumour growth. The bidirectional MRT of 50 µm 211 ctc had the best therapeutic index, while the bidirectional MRT of 25 µm, 211 ctc caused early deaths in 50% of cases four days after irradiation.

Three cross-planar MRT arrays (orthogonal to each other) were used to target 9LGS-bearing rats in three consecutive days at the European Synchrotron [35]. Briefly, day 1 (25 µm; 211 ctc; 400 Gy), day 2 (25 µm; 211 ctc; 360 Gy), and day 3 (25 µm; 211 ctc; 400 Gy). This strategy increased the lifespan of the 9LGS-bearing rats by 216% in comparison to the control group. 

MRT has also been tested on 9LGS in combination with different therapeutic agents. In a study by Régnard and colleagues [36], MRT (25 µm; 200 ctc; 625 Gy peak-dose) was combined with direct intra-tumoral injection of the chemotherapeutic agents’ Temozolomide, Cisplatin, or the contrast agent Gadolinium. Combining Temozolomide and Cisplatin with MRT did not improve survival. However, the concomitant administration of Gadolinium, as a result of radiation dose-enhancement, did. The antimitotic drug Chalcone JAI-51 (an inhibitor of tubulin polymerization) improved survival of 9LGS-bearing rats when combined with two orthogonally cross-fired MRT arrays (50 µm; 200ctc; 480 Gy peak-dose) [37]. Furthermore, Sorafenib (an anti-angiogenic drug) was tested in combination with MRT on the same cancer model [38]. Although the tumour growth rate was similar after MRT and the MRT+Sorafenib groups, there were appreciable changes in the tumour physiology which included reduced tumour blood volume after MRT+Sorafenib.

A study from 2006 shed light on the possible synergistic action between MRT and the immune system [39] where combining MRT (27 µm; 211 ctc; 625 Gy peak-dose) with injections of irradiated 9LGS cells (transfected with the gene GM-CSF), increased the median survival of 9LGS-bearing rats.

Other types of glioma cell lines have also been exposed to MRT. F98 glioma cells and C6 cells, implanted respectively in Wistar and Fisher rats, were used to demonstrate that two orthogonal, cross-planar MRT arrays (24.75 µm; 211 ctc; 350 Gy peak-dose) significantly increased survival as compared to untreated controls [40]. This effect was amplified when MRT was preceded by the administration of the glutathione synthesis inhibitor buthionine-SR-sulfoximine [41]. When two orthogonal MRTs (50 µm; 200 ctc; 241 Gy peak-dose and 10.5 Gy valley-dose) were compared to two orthogonal BB fields (10.5 Gy), MRT was more effective than BB in slowing F98 glioma growth and increasing tumour vessel permeability [42]. The same model of F98 cells in rats was employed by Fernandez-Palomo and colleagues, to prove that the presence of glioma causes different abscopal and bystander effects in MRT-irradiated rats and also in unirradiated (tumour-free) cage-mate rats [43].

C6 glioma-bearing rats were also employed by Fernandez-Palomo et al. for the study of bystander and abscopal effects when comparing MRT with BB [19]; both MRT and BB promoted radiation-induced bystander effects in the non-irradiated portion of the brain and in the bladder of the tumour-bearing animals. This same model has been used to study the dose deposition of MRT (25 μm; 200 ctc; at 35, 70 or 350 Gy peak-dose) using the marker of DNA damage 𝛾-H2AX [44]. Moreover, the damage of the optic nerve was studied after treating the same C6 glioma in Wistar rat with cross-fired MRTs (25 μm; 200 ctc, 350 Gy peak-dose) [45]. The optic nerves did not show any significant histological difference compared to unirradiated controls.

Rat xenografts of glioma cells implanted in the brain of nude mice were also used in MRT studies. In a model of 9LGS cells implanted in nude mice (strain not reported), a significant increase of tumoral apparent diffusion coefficient (ADC), consistent with an increase of vascular permeability, was shown 24 h after the delivery of two orthogonally cross-planar MRT arrays (25 μm; 211 ctc; 500 Gy peak-dose) in irradiated tumours versus unirradiated controls [46]. In addition, athymic nude mice (Crl:NU-Foxn1nu) were implanted with F98 glioma cells to establish that bystander and abscopal effects can also be detected in athymic mice after MRT irradiation (50 μm; 200 ctc; 22 or 110 Gy peak-dose) [47]. 

Finally, it must be mentioned that human xenografts in nude mice have also been explored. In the study performed by Uyama and colleagues [48], U251 human glioma cells, implanted subcutaneously into the flank of nude mice (BALB/cAJc1-nu/nu), were employed to successfully demonstrate the efficacy of one MRT array (100 μm; 500 ctc; 124 Gy peak-dose) and two cross-planar, but unidirectional MRT arrays of (100 μm; 500 ctc; 124 Gy peak-dose) or (20 μm; 100 ctc; 111 Gy peak-dose) at the SPring-8 Synchrotron radiation facility in Japan.

#### 3.4.3. Mammary Tumours

Investigations of the effects of MRT on mammary malignancies have been made at the Japanese and Australian Synchrotrons. The most used pre-clinical model for the study of MRT on mammary malignancies is the subcutaneous injection of EMT-6.5 mouse mammary carcinoma cells in female BALB/c mice. Dilmanian and colleagues [49] injected EMT6.5 cells subcutaneously in the flanks of mice from which tumours were subsequently harvested and small portions transplanted subcutaneously into the calves of BALB/c female mice. These mammary carcinoma-bearing mice were targeted with either one MRT array (90 μm; 300 ctc; doses in Table 1), two cross-planar but unidirectional arrays (90 μm; 300 ctc; doses in Table 1), or with BB of either 23, 30, 38, or 45 Gy. They reported that even one array can control EMT-6.5 tumour growth but that the cross-planar configuration optimally balances tumour control and tissue toxicity. 

By subcutaneously inoculating EMT-6.5 and 67NR mammary tumour cells on the right hind leg of BALB/c mice, Crosbie and colleagues [50] showed a significant increase in median survival compared to controls following one MRT array (25 μm; 200 ctc, 560 Gy). They also showed no signs of tumour re-growth after two different cross-fired MRT arrays (25 μm; 200 ctc; 280 Gy peak-dose) and (25 μm; 200 ctc; 560 Gy peak-dose).

Using the same syngeneic model, a comprehensive transcriptomic analysis of EMT-6.5 tumours, exposed either to MRT (25 μm; 200 ctc; 560 Gy peak-dose) or BB (11 or 22 Gy) compared to non-irradiated tumour controls, underlined how MRT-treated tumours overexpress immune-related genes at early time points after irradiation [51]. Similarly, MRT (25 μm; 200 ctc; 112 Gy or 560 Gy peak-dose) delivered to EMT-6.5 tumours up-regulated genes that are involved in eosinophil recruitment and/or functionality relative to BB (5 or 9 Gy) irradiated tumours [52]. Using the same EMT-6.5 mouse model and same MRT and BB configurations and doses, Yang et al. [53] showed that a different immune response is induced by MRT compared to BB, including a significant increase of tumour-infiltrating T-lymphocytes at 48h post-MRT.

The tissue composition of the EMT6.5 mammary model was investigated after MRT (25 μm; 200 ctc; 560 Gy peak-dose) and BB (11, 22 or 44 Gy) using Fourier-transform infrared microspectroscopy [54]. Absorbance patterns in the nucleic acid region showed chemical shifts between the peak and valley-dose regions [54].

Griffin et al. [55] employed a different type of syngeneic mouse model of mammary carcinoma, where 4T1 cells were injected subcutaneously in the limb of female BALB/c mice. They combined MRT with the anti-angiogenic peptide anginex and showed enhanced tumour growth-delay in combination with MRT.

#### 3.4.4. Melanoma

A recent publication from Potez et al. [7] described for the first time that MRT (50 μm; 200 ctc; 407.6 Gy peak-dose) is an efficient approach for the treatment of radioresistant melanomas compared to a BB (6.2 Gy) irradiation and unirradiated control animals. In this model, B16-F10 melanoma cells were implanted in the ears of C57BL/6J mice.

#### 3.4.5. Squamous Cell Carcinoma

One study from Miura and colleagues [56] employed a murine model of aggressive murine SCCVII squamous cell carcinomas implanted subcutaneously into the left thighs of female C3H mice. They showed that a better palliative effect was elicited by MRT irradiations (35 μm; 200 ctc; 625 Gy or 884 Gy peak-dose and 70 μm; 200 ctc; 442 Gy peak-dose) compared to BB (25 or 35 Gy) irradiations.

#### 3.4.6. MRT for non-neoplastic diseases

Microbeam radiation therapy has also been applied to the treatment of several non-malignant diseases and employed as a tool for studying Central Nervous System (CNS) injury.

##### Epilepsy:

Due to the high tolerance of the central nervous system to MRT, it has been applied, pre-clinically, as a radio-surgical tool to modulate brain networks and treat a variety of neurological disorders including epilepsy. The size and precision of microbeams can mimic a surgical cut to the epileptic cortex reproducing the current surgical ablation techniques but with a reduced risk of inducing neurological dysfunction [57,58]. Microbeam transections of the sensorimotor cortex significantly reduced seizure without inducing neurological dysfunction (100 μm; 400 ctc; 240 or 320 Gy peak-dose) [59]. This is reflected by decreased neuronal excitability of neurons in the somatosensory cortex and reduction of spontaneous synaptic activities and synchronization [60]. Anti-epileptic activity was maintained more than 4 months post-irradiation [61] with an absence of myelin and neurons in the microbeam-irradiated slices [62].

##### Restenosis:

MRT has been implemented as a potential treatment for restenosis following angioplasty, exploiting the vascular-sparing and wound-healing effects of MRT to interfere with the process of cellular hyperplasia, which underlies this condition [63]. The MRT array used in this study (27 μm; 200 ctc; 150 Gy peak-dose) did not yield significant improvement, and future dose-escalation studies are required.

##### Spinal cord Injury:

MRT has also been used as a tool to induce spinal cord injuries to study the mechanisms following CNS injury [64]. Further studies have been performed, but their microbeams width fall outside our inclusion criteria.

#### 3.4.7. Normal Tissue Effects of MRT

It has been well described that in addition to effective tumour control, MRT exhibits exceptional preservation of normal tissues. This tissue-sparing effect is exemplified by the high tolerance of skin, CNS, and vasculature following MRT doses. Table 2 provides a summary of the conservative MRT parameters that each publication cites before the onset of pathology in the majority of test animals. Valley-doses are indicated where provided. 

##### Central Nervous System, Adults:

Slatkin et al. 1995 presented the tissue-sparing effects on normal adult rat brain exposed to MRT (20 μm; 200 ctc) [18]. Brain tissue appeared normal following peak-dose of 625 Gy, necrosis was absent even after peak-doses of up to 5000 Gy, and was visible only after a peak-dose of 10000 Gy. In another study, when multiple arrays were used, brain tissue damage was restricted to the cross-irradiated regions of the two arrays (25 μm; 200 ctc; 625 Gy peak-dose) and normal tissue damage was absent in the brain exposed to only one array [25]. Irradiation of the rat spinal cord did not impair locomotion after 253 Gy (35 μm; 210 ctc), while paresis and foreleg paralysis developed when the dose reached 357 Gy or greater [65]. When valley-doses at the target depth approached 18 Gy (507 Gy peak-dose), white and grey matter necrosis, necrosis of microvasculature and leukocyte infiltration was observed.

##### Developing CNS:

Employing MRT in infant patients was also pursued in pre-clinical studies. Early MRT studies were aimed at assessing its effects on immature brain tissues with suckling rats [26], weanling piglets [66] and unhatched duck embryos [67] as models. In fact, there was a differential dose tolerance of the suckling rat relative to the weanling piglet, with the rat displaying neurological dysfunction when exposed to 150 Gy (28 μm; 105 ctc)[26]. Similarly, irradiation of the CNS of the unhatched duck embryo could withstand a peak-entrance dose of 160 Gy (27 μm; 100 ctc) before displaying signs of ataxia within 75 days of hatching [67]. In contrast, the piglet showed no signs of neurological damage at an dose of 600 Gy (28-30 μm; 200 ctc) [66]. These three studies had comparable beam widths, however, they differed in the beam spacing with 100 μm and 200 μm used in the rat and piglet, respectively. When the spacing was increased to 200 μm, the dose threshold tolerated by the suckling rat increased to 300 Gy without signs of neurological dysfunction [26]. 

##### Eye:

Certain target areas in the brain may result in ocular radiation exposure. The eye of the rat developed retinal degeneration after bidirectional MRT arrays (25 μm; 211 ctc; 350 Gy peak-dose) [68]. This may occur after exposure of the retina to valley doses of up to 14 Gy [69]. The optic nerve was also preserved at these doses [45], however, ocular damage following MRT irradiations to the brain have not been assessed at doses below 350 Gy. Therefore, a tolerance threshold cannot be reported.

##### Vasculature:

The normal tissue response and tissue sparing effect of MRT are dependent on the preservation or regeneration of microvasculature. Vasculature organization of the irradiated tissue depends on the stage of tissue maturation and as a result, determines tolerance to MRT. Models of immature vasculature have been the regenerating fin of the zebrafish [70] and the early chick chorioallantoic membrane [71]. The susceptibility of immature vessels to MRT-induced damage is significantly higher than the mature vessels which show very little post-MRT alterations (vascular lesions, reduced perfusion) when the MRT spacing is kept below 200 µm [71]. Although conducted in non-malignant tissue, the immature vasculature is representative of the disorganized, leaky and fast-growing tumour vessel networks.

In the mouse brain, it has been shown that the microvasculature can withstand up to a 1000 Gy peak entrance dose from a unidirectional MRT array (25 μm; 211 ctc) [72] with only transient blood-brain-barrier leakage. No vascular damage [73], or changes in cerebral water content were observed at this dose [72,74]. In the beam path, there is a progressive disappearance of glial and neuronal cells, but endothelial cells remain detectable in the irradiated tissue and tissue remains perfused. No changes in blood volume or vascular density are observed (up to 3 months post-irradiation) indicating rapid vascular repair [72]. Additionally, peripheral arteries in the mouse can tolerate peak microbeam doses up to 2000 Gy (50 μm; 400 ctc) with minimal damage [75].

##### Skin:

Studies investigating the tolerance of the skin of rats and mice have shown that doses up to 1170 Gy (90 μm; 300 ctc; dose-valley up to 35 Gy) are marked by the absence of moist desquamation, focal denudation or exudation [76]. Moreover, 1740–1900 Gy peak-entry-doses showed significant damage to the skin in both the mouse and the rat with the development of moist desquamation, epilation, and necrosis. The valley-dose was responsible for these adverse reactions, limiting valley-doses to less than 45 Gy for both cross-planar MRT of 2 arrays [49] and a single MRT array [76]. The peak threshold dose is reduced to 650 Gy with a bidirectional microbeam as the volume of irradiated tissue is increased with this irradiation geometry. The skin has been shown to efficiently repair double-strand breaks in the microbeam path (24-84 hrs post-irradiation [pi]) with the absence of apoptosis after MRT (25 μm; 175 ctc) [77]. Doses up to 800 Gy (25 μm; 200 ctc) exhibited similar damage to BB doses of up to 44 Gy [78]. Histological changes following 800 Gy (25 μm or 50 μm; 200 ctc) include thickening of the dermis and epidermis, hyperkeratosis, dermal oedema, and a loss of sebaceous glands [6,78]. 

##### Other tissues:

Lastly, MRT has been applied as a fertility-sparing modality for testicular irradiation and has been able to successfully preserve spermatogenesis in the rat [79]. MRT could, therefore, also be implemented as a treatment modality for testicular cancers or tumours in the inguinal region where infertility is an adverse outcome of current conventional radiotherapies.

##### Importance of Valley-Doses:

Normal brain tissue damage (including radiation-induced necrosis) is highly dependent on the valley-dose, and many authors suggest that cells in these regions repopulate the tissue damage in those regions receiving the peak-dose [25,49,66,73,80,81]. Therefore, it is important to ensure that valley-dose does not exceed the tolerance threshold of the normal tissue [65]. Valley-doses can be reduced either by decreasing the peak-dose, decreasing the width of the microbeam itself or decreasing the beam field-size [34]. Delivered MRT doses attenuate with increasing tissue depth [34], with the peak regions exhibiting a sharper fall-off than the valley regions [82].Ensuring effective dose delivery to the target while maintaining appropriate valley-doses to preserve normal tissue, is therefore an essential step in treatment planning. Many papers in the MRT field do not report predicted doses received by the target tissue, but when provided, these values are reported in Table 2. 

## 4. Discussion

Conventional RT aims to cure cancer by directly ablating tumorous while selectively sparing healthy tissues. All modern RT modalities and techniques, ranging from high energy photons to proton and carbon ions, follow the same paradigm of directly destroying malignant tissue. However, their success is limited by radiation-induced toxicity to surrounding healthy tissue. This is especially critical in radiation-resistant tumours, where high, local doses are required for tumour control. MRT, however, targets tumours and the tumour microenvironment based on a different radiobiological paradigm (spatial fractionation). Thus, even tumour cells not directly irradiated can be eradicated completely while damage to healthy tissue can be repaired efficiently. As a consequence, it is anticipated that MRT will warrant higher tumour control at reduced toxicity in patients.

### 4.1. Potential Underlying Mechanisms of Microbeam Radiation Therapy

#### 4.1.1. Targeted and Non-Targeted Effects of MRT

Peak-doses cause severe radiation damage that is restricted to the cell populations in the path of the microbeam. This can be observed in the brain of rats as early as 8h post MRT (25 μm; 200 ctc; 350 Gy peak-dose) [44], and as late as 128 days post MRT (25 μm; 100 ctc; 625 Gy peak-dose) [25]. In the cerebellum of piglets, microbeam tracks were still visible 15 months post MRT (20–25 μm; 211 ctc; cerebellar peak-dose 66 to 263 Gy; valley-dose 3.2 to 12.6 Gy) [66]. Sharp delimitation of damaged tissue in the microbeam path has also been shown in *Drosophila* tissue where high peak doses exceeding lethal, seamless irradiation doses affect specific morphological processes while maintaining the survival of post-mitotic tissues and the organism as a whole [84].

Radiation-induced bystander effects (RIBE) are relevant for MRT because tissue exposed to the valley-dose will receive signals from neighbouring cells exposed to the peak-dose. Although falling outside of our inclusion criteria due to its employment of a single microbeam and not an array, Dilmanian et al [85] were the first to suggest the importance of RIBE in MRT. Their results from irradiated rat spinal cord indicated that the repair process and the elimination of apoptotic cells in the peak area occurred faster than expected, suggesting that restoration and proliferation was a consequence of “beneficial” bystander factors coming from the valley area. This was further suggested by the results of experiments by Fernandez-Palomo et al [44] performed in the brain of rats. 

When it comes to the effects on non-irradiated tissues (outside of the microbeam array), responses such as genotoxic effects [77] and clonogenic cell death on cells exposed to signals from the irradiated animal [19,43] have been observed after MRT. Some of those responses have involved the immune system [47], with some author suggesting that a functional immune system is key to observe such genotoxic effects post MRT [86].

#### 4.1.2. MRT Selectively Disrupts Immature Blood Vessel

The biological effects induced by MRT go beyond direct tumour cell destruction. In fact, MRT does not impact the morphological and functional characteristics of normal murine brain vessels even after delivery of doses up to 1000 Gy [72]. Brain perfusion, capillary density and blood volume remain unaffected 12h to 3 months after an anteroposterior MRT array (25 μm; 211 ctc; 312 or 1000 Gy peak-entrance dose) [72]. No changes in animal behaviour have been observed [72]. Data from chick chorioallantoic membrane [71] and zebrafish fin regeneration [70] models demonstrate the disruptive vascular effect of MRT on immature blood vessels. Work in adult organisms confirmed that the disruptive vascular effects of MRT depend on the vascular maturation status. In adult zebrafish, a correlation between microbeam width and biological effects of MRT was identified [70]. The study indicated that microbeam spacing between 50 to 100 µm could selectively affect mature and immature vessels. Murine brain vessels do not tolerate beamlets wider than 100µm when peak doses of 400 Gy are delivered [87]. The use of MRT in rodent models revealed a preferential adverse effect on tumour vessels rather than those of healthy tissue. In a murine melanoma model, MRT significantly reduced (24%) the perfusion of the tumour blood vessels indicating vascular disruption [7].

MRT preferentially lowers tumour O_2_ saturation levels in gliosarcoma as a result of reduced endothelial cell density and increased inter-vessel distance following two cross-fired arrays (anteroposterior and lateral; each 50 μm; 200 ctc; 400 Gy peak-entry-dose) leading to tumour hypoxia observed by GLUT-1 overexpression [32,33,38]. However, the persistence of hypoxia may be dose-, time- and tumour-dependant with contrasting evidence of tumour hypoxia in a mammary carcinoma decreasing within 14 days post-irradiation at a dose of 150 Gy [55].

#### 4.1.3. MRT Transiently Increases Tumour Blood Vessel Permeability

Few reports indicate that MRT causes a partial disintegration of the endothelium leading to an early (1 to 4h) but significant increase in tumour blood vessel permeability (manuscript in preparation). This “MRT-induced vascular permeability window” represents a potent drug delivery strategy. Combination of MRT with chemotherapy or nanoparticles applied in the “permeability window” results in significantly better tumour control (manuscript in preparation). A recently published study indicated that MRT induced selective vascular permeability in tumour blood vessels but not in normal vasculature. The tumour vascular permeability was observed from days 2 to 7 after MRT. In contrast, it was observed only at day 7 after BB radiotherapy [42]. This study also documented that delivering a second MRT array induced earlier, more pronounced, and more persistent tumour vascular permeability than BB.

#### 4.1.4. MRT Boosts the Recruitment of Circulating Immune Cells into Tumours 

Data obtained in zebrafish indicate that MRT triggers an acute inflammatory response restricted to the regenerating tissue [70]. Six hours post-MRT, the regenerating tissue was infiltrated by neutrophils, and thrombocytes which adhered to the cell wall locally in the beam path. The mature tissue was not affected by microbeam irradiation [70]. In a mouse tumour model, it has been demonstrated that BB markedly increases tumour-associated macrophages and neutrophils while there were no increases in these populations following MRT [53]. The role of MRT in the recruitment of immune cells represents a promising immunoprophylactic treatment strategy and should be further explored in experimental and clinical studies. A recently performed study in melanoma-bearing animals showed that MRT (particularly after 2 irradiations) selectively increased production of natural killer (NK) cell- and cytotoxic T lymphocytes-attracting chemokines while, also showing, decreased secretion of T regulatory cell-, tumour associated macrophage- and neutrophil-recruiting chemokines (manuscript in review). The role of single and repeated MRT administration in the recruitment of immune cells represents a promising immunoprophylactic treatment strategy and should be further explored in experimental and clinical studies. 

In addition, increased pericyte density following MRT suggests a vascular normalization effect [55] which may increase migration of immune cell populations to the tumour and modulate cytokine expression that may support an anti-tumour immune response. Synergism between immunoprophylaxis and MRT has been documented [39]. Differential modulation of genes involved in inflammation and immunity are also marked at early timepoints between tumour and normal tissue [29,52] and increased leukocyte infiltration into the tumour has also been observed. 

### 4.2. Clinical Translation

Barriers to the translation of MRT relate to a limited understanding of how to compare the radiobiological effects of MRT with conventional BB radiotherapy, as well as, understanding the biology behind the physical properties of the microbeams. In addition, the optimal therapeutic strategy for MRT is still to be defined.

#### 4.2.1. Comparative Therapeutic Effects of MRT and BB Radiotherapy

A limited number of studies systematically compared the biological effects of MRT with BB radiotherapy. The superiority of MRT compared to synchrotron BB treatment for tumour control has been demonstrated in mice bearing B16-F10 melanoma [7] and rats bearing 9LGS [31]. In these studies, the MRT valley-dose was kept equal to the BB dose; however, future studies could use the Equivalent Uniform Dose concept (EUD) to determine more biologically comparable doses of MRT and BB [88].

Dose-escalation studies in healthy rodents have identified doses of MRT and BB predicted to cause equal rates of late spinal cord myelopathy [65], lethal neurotoxicity [80] and acute toxicity following total and partial body irradiation [83]. Of these studies, only Smyth et al. [83] included a comparison between MRT and BB radiotherapy at a conventional dose-rate (in the order of 2 Gy min^−1^). This indicates the need for further studies controlling for the potential FLASH tissue-sparing effects of synchrotron broad-beam irradiation.

#### 4.2.2. Identifying Optimal Targets and Strategies for MRT

A key limitation of MRT is the need for a kilovoltage energy spectrum to maintain spatial fractionation on a microscopic scale, leading to 30–40% of the dose being attenuated within the first few centimetres of tissue [89,90]. In addition, the PVDR, which is a metric intrinsically linked to the therapeutic ratio of MRT, decreases with increasing tissue depth and field size [89]. The majority of pre-clinical studies to date have been performed in rodent models (Figure 3), where this issue of beam penetration and PVDR is less important due to the small dimensions of the target. However, human diseases will require larger field sizes and target depths [91].

Given that PVDRs are likely to be lower for human treatments [91], a strategy which relies on relatively low peak-dose (100–250 Gy) by leveraging the immunomodulatory [29,53] or angio-disruptive effects of MRT [42,70] could strike an optimal balance between toxicity and tumour control. Previous studies have shown MRT to be effective for treating rat models of glioma when combined with chemotherapeutic agents [36,37,41] or immunotherapy [39]. Future studies should investigate the optimal dose, sequencing and timing of MRT in relation to chemotherapy and immunotherapy and whether there is a role for temporally fractionating the delivery of MRT. This is particularly relevant for combined radio-immunotherapy, where fractionated radiotherapy has been shown to be more effective than a large single fraction for stimulating an immunogenic response [92,93]. The combination of MRT with immunotherapy is an under-explored frontier which warrants the attention of future pre-clinical studies. A previously established model of metastatic melanoma [7] could be ideal in this context, given the clinical use of radio-immunotherapy for melanoma.

#### 4.2.3. Candidate Animal Models

Animal models for future research should reflect technically feasible and medically justifiable clinical scenarios for a human trial of MRT. Although various brain malignancies have been the major focus of pre-clinical MRT research and proposed as future targets for clinical MRT [94,95,96], the depth of intracranial diseases, the need for beams to transverse the skull, and the potential proximity to sensitive structures could make it a difficult target for the first clinical trials. However, lessons can be learned from the relatively newer field of FLASH radiotherapy which has already been translated to the clinic with the first human FLASH treatment being for cutaneous T-cell lymphoma on the forearm [22].

Similarly, superficial or cutaneous lesions, as well as those of the appendicular musculoskeletal system [96], would be ideal candidates for the first human trials of MRT. Therefore, future animal models for MRT research could include murine models of cutaneous T-cell lymphoma [97,98,99], locally recurrent breast cancer [100], soft-tissue sarcoma [101,102] and osteosarcoma [103,104,105]. These are diseases which can be highly radio-resistant and are often refractory to conventional radiotherapy. Ideally, murine models should be immunocompetent in order to explore and harness the immunomodulatory effects of MRT. Furthermore, soft-tissue sarcoma and osteosarcoma occur spontaneously in canines and are well-established models of human disease [106,107]. Veterinary patients would be on a size and scale more comparable to humans and could provide an intermediate step towards a clinical trial.

### 4.3. Limitations of this Study

A limitation of this study was that some articles, whose sole purpose was studying the effects of MRT, fell outside our inclusion criteria because the researchers used a single beam instead of an array. We would like to report that in one case (properly informed in the text of this scoping review), we cited one of these articles to complement other animal studies in MRT. 

## 5. Conclusion and Recommendations

### 5.1. Appropriate Controls

Historically the dose delivered by a synchrotron BB has been considered as a control for MRT. However, BB originating from synchrotron sources has the potential of achieving FLASH normal tissue sparing effect. Therefore, the therapeutic advantages of MRT over BB may have been substantially underestimated in these studies, and appropriate BB irradiation controls, using conventional dose-rates from clinical linear accelerators or conventional kilovoltage X-ray sources should be used. Additionally, comparative doses should be based on biological equivalence, such as Equivalent Uniform Dose, rather than valley-dose alone.

### 5.2. Standardization of the MRT Array and Doses

Amongst the literature, there is a wide variation in the parameters used for the MRT array and peak and valley-doses. We recommend working on establishing a method for tracking and standardizing the effects of these variations. We hope that the two tables presented in this scoping review can serve as a starting point when selecting radiation parameters for future studies. So that effective tumour control with effective normal tissue sparing can be achieved. Moreover, progress is needed to identify the optimal MRT array and doses required for adequate vascular disruption, vascular permeability and recruitment of tumour-infiltrating immune cells in each animal model. 

### 5.3. Preclinical Animal Models

The most employed tumour models, especially for brain tumours, have been established in rats. We recommend diversifying the use of animal species and including new tumour cancer models to accelerate the clinical translation of MRT.

### 5.4. Veterinary Trials

Veterinary trials, in canines or felines, will provide an important intermediary step between small, pre-clinical animal models and human treatments. These trials will be crucial to ensure the radiobiology and physical characteristics of MRT can be translated to humans despite the challenges of size and beam attenuation. Furthermore, large animal models will validate a clinical workflow for MRT, including custom dose-calculation algorithms, conformal treatment planning capabilities, image-guidance, patient-positioning systems, and reporting and verification of dose coverage of the target. At this stage, superficial lesions of extremities are the likely first candidates for veterinary trials, considering their current use in translational research, their prevalence in the canine population and a favourable anatomic location and depth for MRT.

We believe that the consolidation of these five recommendations may result in a significant advance and rapid adoption of MRT in the clinic.

## Figures and Tables

**Figure 1 cancers-12-00527-f001:**
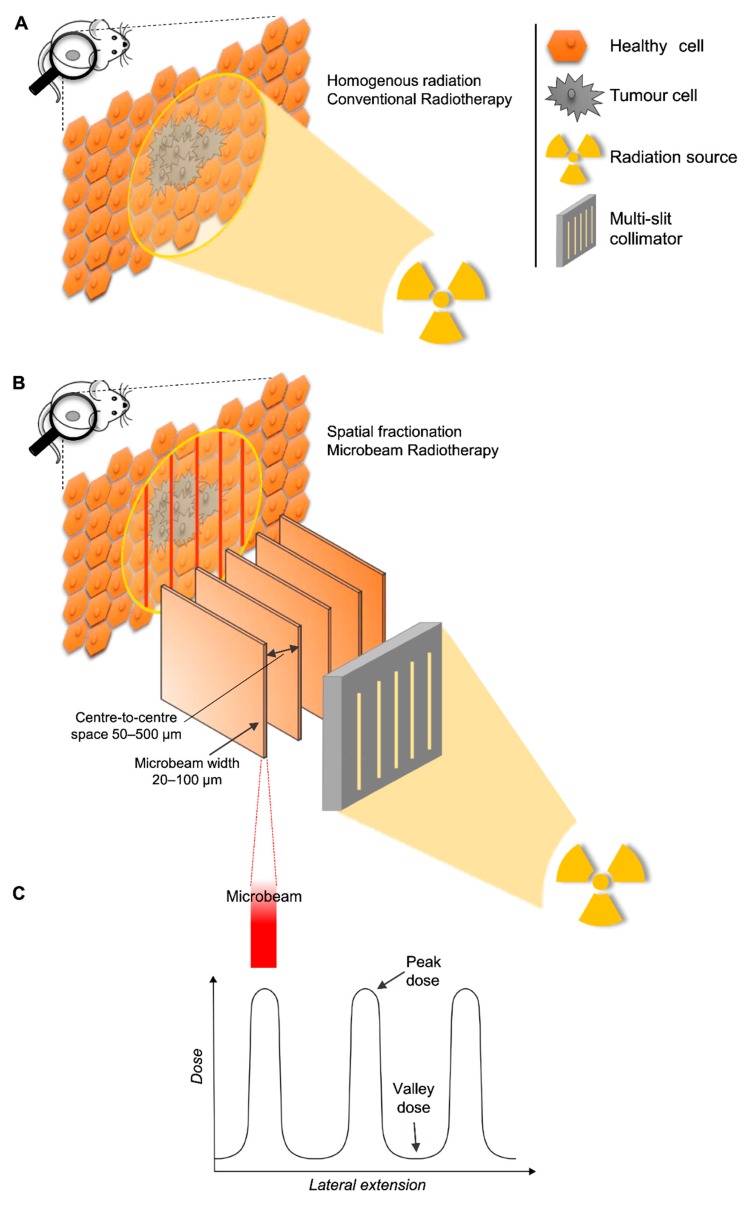
Homogenous radiation vs spatial fractionation. Schematic representation of a homogenous beam of radiation (panel **A**) and a MRT array (panel **B**) targeting a malignancy. Schematic representation of the microbeams dose distribution (panel **C**).

**Figure 2 cancers-12-00527-f002:**
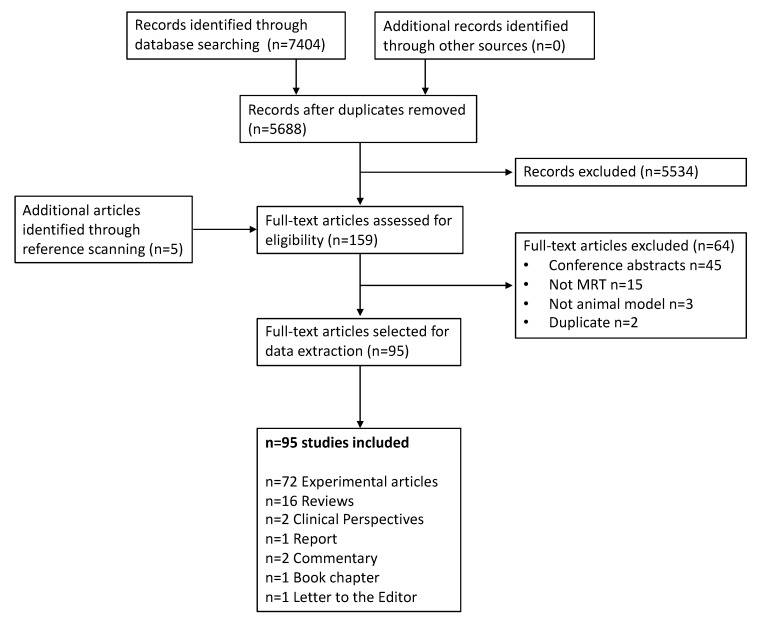
Study Flow. Details of the flow of information during the study.

**Figure 3 cancers-12-00527-f003:**
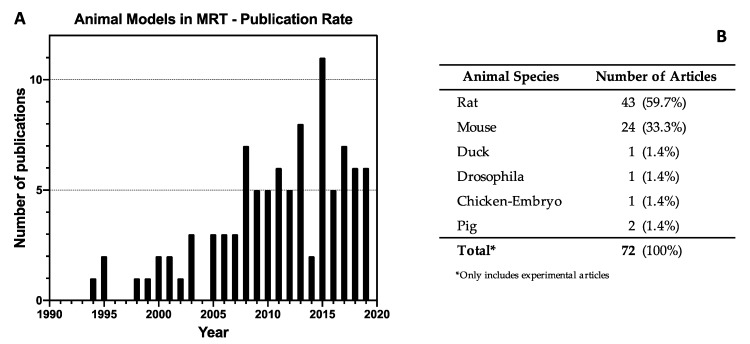
Publication rate and frequency of animal species used as animal models in Microbeam Radiation Therapy (MRT). (**A**) Shows the yearly publication of full-text articles that described Animal Models in MRT (including reviews and commentaries). (**B**) Shows the word count and frequency of all animal species used (it only includes the 72 experimental articles).

**Table 1 cancers-12-00527-t001:** MRT Parameters Used in Cancer Models.

Animal	Cancer Type	Number of Arrays	Microbeam		Peak-dose (Gy)	Valley-dose (Gy)	Evaluated Criteria
			Width (µm)	Spacing (µm)			
Rat [1]	Gliosarcoma (9L)	1 & 2	50	200	400 entrance dose; 350 @1cm depth	12.5 @1cm depth	Animal survival, tumour growth, tumour vasculature, and cell proliferation
Rat [2]	Gliosarcoma (9L)	2	50	200	480 entrance dose; 418 @1cm depth	18.6 @1cm depth	Animal survival, cell cycle, and DNA distribution patterns
Rat [3]	Gliosarcoma (9L)	2	50	200	400 entrance dose; 350 @1cm depth	12.5 @1cm depth	Tumour vasculature and tumour oxygenation
Rat [4]	Gliosarcoma (9L)	1	50	200	400 dose @tumour (i.e., @7mm depth)	18 dose @tumour (i.e., @7mm depth)	Animal survival and Transcriptomics
Rat [5]	Gliosarcoma (9L)	1	50	200	400 dose @tumour (i.e., @7mm depth)	8 dose @tumour (i.e., @7mm depth)	Tumour growth, transcriptomics, and histopathology
Rat [6]	Gliosarcoma (9L)	1	50	200	400 dose @tumour (i.e., @7mm depth)	17.4 dose @tumour (i.e., @7mm depth)	Animal survival, tumour growth, cell proliferation, and gene expression
					200 dose @tumour (i.e., @7mm depth)	8.7 dose @tumour (i.e., @7mm depth)	
Rat [7]	Glioma (F98)	2	50	200	241.4 entrance dose	10.5 @9mm depth	Tumour vasculature and tumour oxygenation
Mouse [8]	Mammary (EMT6.5/67NR)	1	25	200	560 entrance dose	8.5 @centre of brain	Animal survival, DNA damage, cell proliferation, and apoptosis
					800 entrance dose	12 @centre of brain	
		2			280 entrance dose	8.5 @centre of brain	
					560 entrance dose	17 @centre of brain	
Rat [9]	Gliosarcoma (9L)	1	27	50	150 entrance dose; 108 @centre of brain	20 @centre of brain	Animal survival and histopathology
					250 entrance dose; 179 @centre of brain	34 @centre of brain	
					300 entrance dose; 215 @centre of brain	40 @centre of brain	
				75	250 entrance dose; 179 @centre of brain	17 @centre of brain	
					300 entrance dose; 215 @centre of brain	20 @centre of brain	
					500 entrance dose; 359 @centre of brain	33 @centre of brain	
				100	500 entrance dose; 359 @centre of brain	19 @centre of brain	
Mouse [10]	Mammary (EMT6.5)	1	90	300	800 dose @tumour	16 dose @tumour	Tumour ablation
					890 dose @tumour	18 dose @tumour	
					970 dose @tumour	19 dose @tumour	
					1740 dose @tumour	35 dose @tumour	
					1820 dose @tumour	36 dose @tumour	
					1900 dose @tumour	38 dose @tumour	
		2	90	300	410 dose @tumour	16 dose @tumour	
					520 dose @tumour	21 dose @tumour	
					650 dose @tumour	26 dose @tumour	
Rat [11]	Glioma (C6)	1	25	200	17.5, 35, 70, 350 entrance dose	0.51, 1.03, 2, 10.3	Bystander effects in-vivo by clonogenic cell survival
Rat [12]	Glioma (C6)	1	25	200	35, 70, 350 entrance dose	NR	DNA damage
Rat [13]	Glioma (F98)	1	25	200	20, 200 entrance dose	NR	Bystander effects in-vivo by clonogenic cell survival and cellular calcium fluxes
Mouse nude [14]	Glioma (F98)		50	400	22, 110 entrance dose	0.5, 2.5	Bystander effects in-vivo by clonogenic cell survival and cellular calcium fluxes
Mouse [15]	Mammary (4T1)	1	50	200	150 @5 mm depth	7.5 in a 10 mm solid water phantom	Tumour growth, tumour vasculature, and tumour hypoxia
Mouse [16]	Mammary (EMT6.5)	1	25	200	112, 560	NR	Immune response by gene expression and histopathology
Rat [17]	Gliosarcoma (9L)	1	50	200	400 entrance dose	NR	Tumour vasculature, and tumour hypoxia
Rat [18]	Gliosarcoma (9L)	1	25	100	625 entrance dose	NR	Animal survival, tumour growth, and histopathology
Mouse [19]	Squamous cell carcinoma (SCCVII)	1	35	200	442, 625, 884 entrance dose	NR	Animal survival, tumour growth, and tumour ablation
			70	200	442 entrance dose		
Rat [20]	Glioma (C6)	2	25	200	350 entrance dose	NR	Optic nerve damage by histopathology
Mouse [21]	Melanoma (B16F10)	1	50	200	407.6 dose @tumour	6.2 dose @tumour	Tumour growth, tumour vasculature, cell proliferation, cell senescence, and immune response
Rat [22]	Gliosarcoma (9L)	1	25	200	625 entrance dose	12.1 dose @tumour	Animal survival, tumour growth, and histopathology
				100	625 entrance dose	36 dose @tumour	
Rat [23]	Gliosarcoma (9L)	1	25	200	625 entrance dose	NR	Animal survival, tumour growth, and histopathology
Mouse [24]	Mammary (EMT6.5)	1	25	200	560 entrance dose	11	Biochemical changes by synchrotron Fourier-transform infrared microspectroscopy
Rat [25]	Glioma (C6, F98)	2	25	211	350 entrance dose	NR	Animal survival and object recognition
Rat [26]	Glioma (F98)	2	28	400	350	18 dose @ tumour	Animal survival and cognitive dysfunction
Mouse nude [27]	Gliosarcoma (9L)	1	25	211	500 entrance dose	24 (cross-fired)	Animal survival, tumour growth, and tumour vasculature
Rat [28]	Gliosarcoma (9L)	2	25	211	860 entrance dose	18 @1cm depth	Animal survival and histopathology
			50		480 entrance dose		
			75		320 entrance dose		
Rat [29]	Gliosarcoma (9L)	3	50	211	400, 360 (+24h), 400 (+48h) entrance dose	15	Animal survival and histopathology
Rat [30]	Gliosarcoma (9L)	1	27	211	625 entrance dose	NR	Animal survival, histopathology, and immune response
Mouse [31]	Mammary (EMT6.5)	1	25	200	560	11	Transcriptomics
Mouse nude [32]	Glioma (U251)	1	100	500	124	4.8	Tumour growth, histopathology, and apoptosis
		2	20	100	111	8.2	
			100	500	124	9.6	
Mouse [33]	Mammary (EMT6.5)	1	25	200	112, 560	NR	Immune response

NR: Not-Reported.

**Table 2 cancers-12-00527-t002:** MRT Parameters Eliciting Normal Tissue Tolerance.

Animal	Tissue	Microbeam		Pathology	Threshold Peak-Dose (Gy)	Valley-Dose (Gy)
		Width (µm)	Spacing (µm)			
Duck [67]	Immature CNS	27	100	Ataxia	160	NR
Rat [15]	Brain	27	75	Necrosis, oedema	250	17
		27	100		500	33
Rat [73]	Brain	27	200	Cell loss, demyelination	1000	NR
Rat [25]	Brain	25	100	Tissue damage (loss of tissue structure; vascular damage)	625 uni- & bidirectional	NR
					312 bidirectional	
Rat [65]	Spinal cord	35	210	Paresis/Paralysis (over 383 dpi)	357 (19mm transverse depth)	12.7
Mouse [72]	Brain	25	211	Damage to microvasculature/vasogenic oedema (up to 1 mpi)	312–1000	5.8
Rat (11–13 day old) [26]	Brain	28	105	Neurological dysfunction	150	>5 Gy [69]
		28	105		50	
		25	210		300 bidirectional	
Piglet [66]	Brain	20–30	210	Neurological function (465 dpi)	600 peak-entry 263 @ cerebellum	12.6 @ cerebellum
Rat [18]	Brain	20	200	Necrosis (14 dpi)	5000	NR
		37	75	Tissue damage (14–31dpi)	625	
		20	200	Loss of nuclei (30-31dpi)	2000	
		37	75			
	Skin	37	200	Epilation (2–4 weeks dpi)	1250–2500	
		37	75		1250–2500	
Mouse [74]	Brain	25	211	Cerebral oedema (up to 28 dpi)	<1000	10.5
Mouse [49]	Skin	90	300	Moist desquamation	1740 co-planar	35
					650 cross-planar	26
Mouse [76]	Skin	90	300	Moist desquamation	1005-1170	14-17
Mouse [83]	Abdomen	25	400	GI syndrome	249 (257 TD_50_)	7.5 (7.7)
	Head			Neurological dysfunction	255 (268 TD_50_)	6.8 (7.2)
	Thorax			Neurological dysfunction, pulmonary damage	391 (lowest administered dose)	9.5
	Total body			Weight loss, moribund behaviour	88.9 (120 TD_50_)	2.8 (3.8)
Mouse [75]	Saphenous artery	50	400	Functional deficits, atrophy, arterial damage	2000	17.6
Rat [34]	Brain	50	211	Neurological dysfunction	480	18
Mouse [79]	Testes (ex vivo)	50	100	Spermatogenesis	5	NR
Rat [68]	Brain (+eye)	25	211	Retinal degeneration (12 dpi)	350 * (bidirectional)	NR
Rat [45]	Brain (+eye)	25	200	Optic nerve damage	350 * (bidirectional)	NR

Threshold Peak-Dose: indicates the highest peak-dose (skin entry or at indicated tissue depth) reported that did not result in pathology in the majority of animals. Note: studies testing only one dose are indicated by an asterisk (*). NR: Not-Reported.

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
