# Peer review of "Animal Models in Microbeam Radiation Therapy: A Scoping Review"

_cancers, 2020, doi:10.3390/cancers12030527_

Round 1
Reviewer 1 Report
The manuscript „Animal Models in Microbeam Radiation Therapy“ summarizes the preclinical in-vivo work in microbeam radiation therapy. In my opinion this is a very valuable and important contribution and of high interest for the microbeam community. Moreover, the manuscript highlights many challenges and problems in preclinical work with microbeams – such as a lack of published dose and beam geometry parameters.
Some minor comments:
- Abstract, line 21: “… according to their use in cancer treatment …” (used --> use)
- Methods, line 64: PROSPERO should be explained
- Is Section 2.6 necessary? I would remove it.
- Small/capital letters: The should be a convention throughout the manuscript on how to write certain terms – e.g. spatial fractionation or Spatial Fractionation. (I suggest to use small letters)
- Figure 3:
- I suggest to only present the table
- Should pig and piglet be classified as separate species?
- Results, line 165: Should co-planar not refer to two or more fields, where microbeam (planes) are parallel to each other? (otherwise two beams are always co-planar, because they are in a plane with each other) To me the definitions provided remain unclear. Consider a revision.
- Results, line 181: FLASH effect not properly defined. (Wording problem)
- Results, line 234: I would skip the 2x.
- Results, line 282: “… that even one array …”
- Results, line 286: “Thy also showed no signs of …”
- Section 3.4.2:
- As the majority of the review deals with MRT in cancer treatment, this section should be removed
- Otherwise, the animal models (e.g. in epilepsy treatment) would require some deeper reflections.
- Results, line 346: “… MRT doses that greatly exceeded those of conventional radiotherapy ... “
- I would be careful with this statement. As the authors describe later on, it is difficult to find an adequate measure to compare conventional and MRT doses. Probably valley dose is much more relevant when comparing BB and MRT. In this aspect also the “exceptional preservation of normal tissue” should be interpreted.
- Also in the entire manuscript conclusions with respect to peak dose should be state with care (e.g. line 408/409)
- Results, line 356: “… spinal cord at 19 mm depth …” ambiguous wording.
- Results, line 362: “Due to the limitation in radiation therapy on infant patients, …” improve wording
- Results, line 422: I think there are 3 ways to reduce the valley dose: reduce peak dose, reduce field size, increase the spacing to beam width ratio.
- Results, line 425-427: … but at the time they show in this paper, that small beams with a very high peak dose are not as good as wider beams with a moderate peak dose. I think this is not the right conclusion from this paper.
- Discussion, lines 495-498: improve wording. These sentences remain vague.
- Conclusion, line 587: Linacs are probably not suitable, as they have a completely different radiation quality. Comparability is therefore questionable.
General comment: What I am missing in the manuscript is a review on the end-points (e.g. tumour growth delay, tumour control, early/late side effects etc.). Although some end-points are discussed individually in the text, it would be worth to include them even in Table 1. It would be interesting to evaluate, whether studies come to similar or even contradicting conclusions.
Author Response
We would like to thank the reviewer for the positive feedback, for taking the time to read the manuscript, and for providing helpful advice. We hope that we have addressed all the comments adequately. Please see our answers the attached document.

Reviewer 2 Report
The authors summarized their review of about 100 papers in the field of animal models in microbeam radiation therapy. The made very useful recommendations for the advancement and rapid adoption of MRT in the clinic. The review is very well written and provides a comprehensive understanding of the MRT and its application as well as the normal tissue toxicity. I recommend the acceptance of its publication after correcting some typos I found:
Line 173, there should be a space between value and Gy. There are also many others in the whole paper.
Line 255, it should be gamma (symbol) –H2AX, not y
Line 592, it should be establishing
Author Response
We would like to thank the reviewer for the positive feedback and for taking the time to read the manuscript. We have corrected the typos and we thank you for identifying them.
